# Parameter δ¹⁸O in the Marine Environment Ecosystem Studies on the Example of the Barents Sea

**A. A. Namyatov \*, P. R. Makarevich, E. I. Druzhkova and I. A. Pastukhov**

Murmansk Marine Biological Institute, Russian Academy of Science, 183010 Murmansk, Russia
\* Correspondence: alexey.namyatov.a@gmail.com; Tel.: +7-7921-9042866

**Abstract:** The isotopic parameter δ¹⁸O in oceanography is used for the calculation of mixing proportions of the Atlantic, rivers, and melted waters as well as the relative content of ice-forming waters. Using these values, as well as nutrient concentrations in the nuclei of these waters, it is possible to calculate a conservative concentration, which is determined only by the water's mixing. These values are the points of reference for water nutrients reserve changes at the expense of the «nonconservative» factors (photosynthesize and geochemical sedimentation). This approach in the calculation of primary production allows moving from the use of the constant stoichiometric ratios to the floating ones, which is observed in the actual ecosystem. Based on the proposed method, the nutrient consumption features and production changes in the Barents Sea were studied. According to the maximum value of production in the summer, the following areas were identified—an early autumn period in the Barents Sea, an area with the maximum values of gross primary production (GPP > 150 g C m⁻²), three regions with increased values (GPP > 100 g C m⁻²), and two regions with relatively low values (GPP < 100 g C m⁻²). The use of this technique with a long-term data series available on salinity and the content of nutrients will allow the future to proceed to the study of the climatic variability of these parameters, ranging from the nutrient consumption variability to the productivity variability of the study area.

**Keywords:** Barents Sea; Svalbard archipelago; nutrients; phytoplankton; primary production; net primary production; gross primary production; photosynthesis

## 1. Introduction

The waters of the Barents Sea are highly productive water areas, which account for a significant amount of primary production on the pan-Arctic shelf [1,2]. Due to its geographical location, and the formation of terrestrial and marine ecosystem features, the Barents Sea is a unique testing ground for studying the entire natural environment responses spectrum of the high-latitude Arctic to external impacts of natural and anthropogenic origin. One of the tasks in such studies is to identify and study areas that are potentially most favorable in terms of the food supply state for fish and other commercial objects, i.e., areas with the highest supply of phytoplankton with nutrients. It is a phytoplankton that plays a decisive role in the formation of the food base of various aquatic organisms, including commercial fish [3]. At the present stage, there are several approaches to assessing phytoplankton production:

- The first approach is the direct measurements in the summer period. They are presented by conventional methods [4,5] such as oxygen and radiocarbon [3].
- The second approach is based on the calculation of chlorophyll, both directly measured in seawater samples and obtained from Earth's remote sensing data [6].
- The third is model calculations, including both hydrodynamic and ecosystem models [6].
- The fourth involves using traps for solid particles, followed by the removal of inorganic carbon [7].

- The next approach is based on determining nutrient intake and calculating primary production using average stoichiometric ratios. This method is based on research carried out in the last century. The advantage of this approach is that the methods of chemical analysis for the determination of nutrients are quite simple, unified in different countries, and well-developed for their use in the field. In addition, large databases have been accumulated. They contain such values for various areas of the World Ocean.

The method for primary production calculation based on nutrient intake estimation is derived from studies carried out in the last century. In the works of A.P. Vinogradov [8,9], R. Fleming [10], and L. Cooper [11,12], the values of the content of phosphorus and nitrogen in phytoplankton and water are presented. A summary of these studies is presented in the monograph—Ocean Chemistry [13].

When calculating the primary production by carbon mass, as a result of the consumption of nutrients, stoichiometric ratios are used. Usually, Redfield–Richards stoichiometric ratios are used, which in the molar form are C:Si:N:P = 106:23:16:1 [1,6,14].

Usually, few forms of primary production are defined [15]. Gross primary production (GPP) is defined as photosynthesis that does not take into account the simultaneous respiration of algae or the metabolism of heterotrophic organisms. Net Primary Production (NPP) is GPP minus algae respiration. Net community production (NCP) is defined as GPP minus autotrophic and heterotrophic respiration.

The regeneration of silicon, in comparison with nitrogen and phosphorus, is extremely slow; as a result, the smallest values of production, as a rule, are equivalent to the loss in the euphotic layer of nitrogen and phosphorus, and the largest—to silicon. Moreover, when assessing the production of phytoplankton by changing the content of silicon, the part of the production that is synthesized from the recycling of nitrogen and phosphorus is also taken into account [2,16,17]. Nitrates support primary production (new production) and are thought to represent the uptake of nutrients present at the start of the growing season or are carried into the euphotic zone from elsewhere.

Summing up the presented analysis following the approach of Arzhanova N.V. et al. [16,17], the NCP calculated silicon production (NCP$_{Si}$) reflects the total production, while the nitrogen calculation (NCP$_{N}$) gives the value of "new" production. In these works, the definition of "new" production corresponds to the definition of "clean" production [1]. In this paper, the same definitions and notation are used.

The current methods for determining biological productivity, based on changes in the concentrations of nutrients, are based on determining the difference between the measured amount of a nutrient and its amount, which "was in a given volume of water at the time of its physicochemical properties formation in the surface layer of the ocean" [18].

In some works, these quantities are called preformed concentrations: preformed—phosphates (P$_{pf}$), preformed—nitrates (N$_{pf}$), and preformed—silicates (Si$_{pf}$). The calculation of preformed concentrations is based on the application of stoichiometric coefficients for phosphorus, nitrogen, and silicon to oxygen and carbon [2,16–18]. The preformed concentration is the reference point for the content of the nutrients (maximum concentration), from which the decrease in this value is counted in the process of photosynthesis. This method was used to assess the biological productivity of various water areas of the World Ocean –of the Bering Sea, [16–18], Antarctic water [19], the Sea of Okhotsk [2], and the Barents Sea [20,21]. To use this technique, it is necessary to know the winter concentrations of nutrients preceding the spring phytoplankton bloom. In works on the Bering Sea, these concentrations were determined by the concentrations at the lower boundary of the cold intermediate layer, the position of which varied from 40 to 170 m. In works on water areas surrounding Antarctica, this concentration was determined as the weighted average value in the layer of autumn–winter convective mixing. Such approaches are not always applicable for several reasons.

In the case when the change in the concentration of nutrients along the vertical is caused not only by the process of photosynthesis but also by the presence in the surface

layer of another water mass, with a different initial composition of nutrients, which is typical for the southeastern part of the Barents Sea, as well as for the Kara and Laptev Seas.

Changes in oxygen concentrations are caused not only by photosynthesis processes but also by exchange with other layers of water and with the atmosphere.

For the transition from the value of nutrient consumption to primary production, i.e., to carbon consumption, current average stoichiometric Redfield–Richards ratios are used; but the use of these ratios for phytoplankton with different species composition can lead to large errors since the ratios of the elements presented differ significantly in different systematic groups; for example, the C:P ratio in diatoms and dinoflagellates differs by 37%, the C:N ratio by 24%, and C:Si by 15 times [13].

The difference shown above is determined not only by the "productive" but also by the "non-productive" components, which may depend on the element under study input from the bottom layers, as well as on the contribution of other processes.

To avoid the above disadvantages, in this paper we propose to use the isotope parameter $\delta^{18}O$, which has unique conservative properties. It does not depend on chemical–biological processes and is an ideal tracer for balance estimates. The use of this parameter makes it possible to solve one of the main problems in the assessment of primary production using changes in the concentration of nutrients—to determine the starting point, i.e., the maximum concentration of a nutrients before the start of the photosynthesis process. Determination of the difference between the reference point and the measured nutrient concentration, i.e., consumption values, allows us to move from average stoichiometric ratios to real ones—corresponding to the real composition of the systematic groups of phytoplankton.

In this paper, we present in detail the methodology for calculating primary production using the isotope tracer $\delta^{18}O$. The results of testing this technique in the Barents Sea are presented. For the first time, a comparison of the estimated contributions of diatoms and dinoflagellates to primary production with the values of direct determination of phytoplankton biomass is shown. Verification of calculations of primary production, performed by other methods and presented by other teams of authors, was carried out.

## 2. Materials and Methods

The presented methodology consists of two blocks. The hydrological block presents an estimate of the base waters value in terms of salinity and stable isotope. In this case, the parameter $\delta^{18}O$ is used. The ecosystem block represents the calculation of primary production based on the base water values and the change in the values of nutrients. For both hydrological and ecosystem blocks, data from the oceanographic data bank National Oceanographic Data Committee (NODC) and World Ocean Atlas 2018 (WOA18) were used (Figure 1). These are the values of salinity, phosphorus–phosphate, nitrogen–nitrate, and silicon. The data in this atlas are presented as dataset grids, in 1° increment in both latitude and longitude, at standard horizons from 0 till 100 at step 5 m and from 100 to the bottom layer at step 25 m. In this work, we used the average monthly values of the parameters above [22].

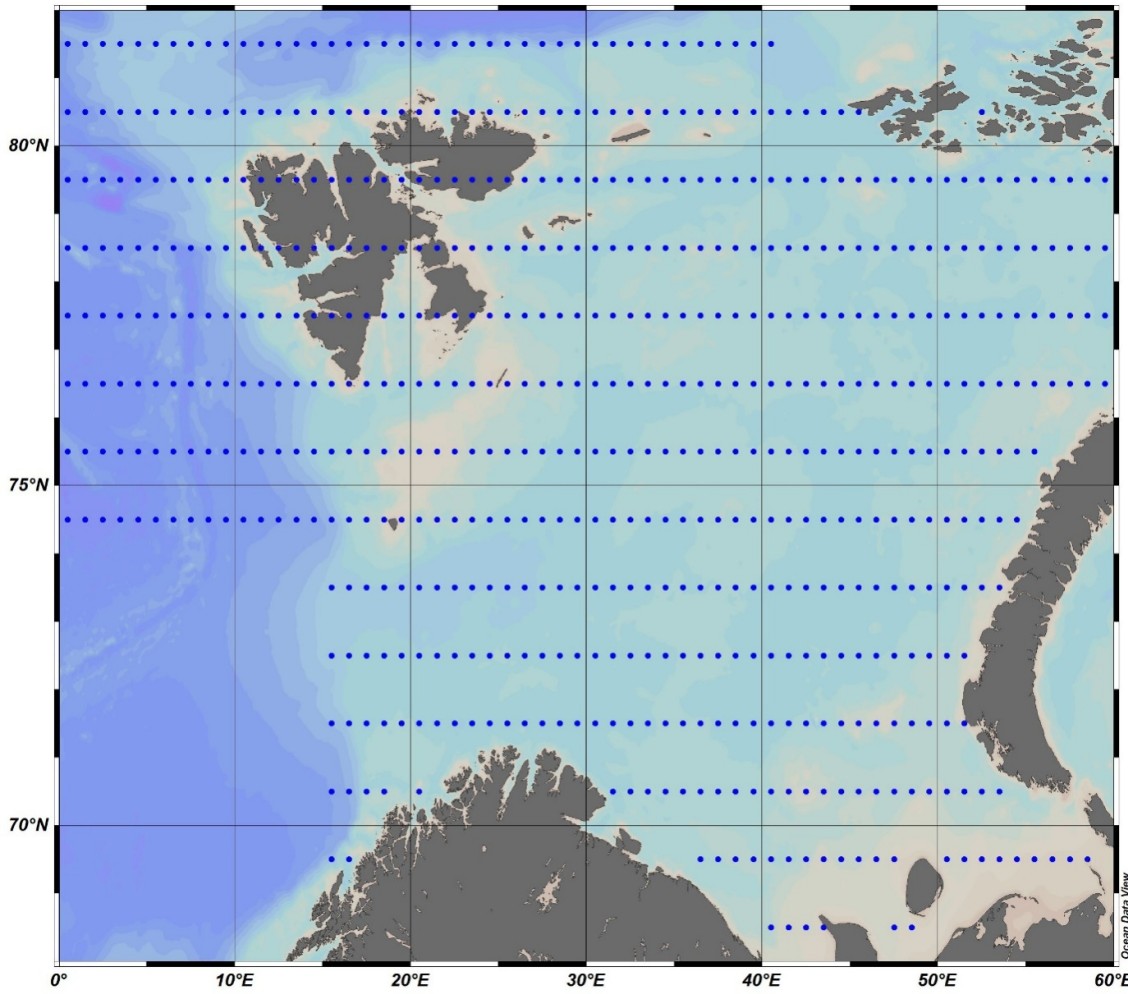

**Figure 1.** Position of grid nodes (blue points) in the oceanographic atlas WOA18 [22].

*2.1. Principle Scheme of Primary Production (PP) Calculation (Ecosystem Block)*

The PP calculation scheme is presented here for the first time. In each grid point, the balance of any nutrients at a point is shown in (Figure 2).

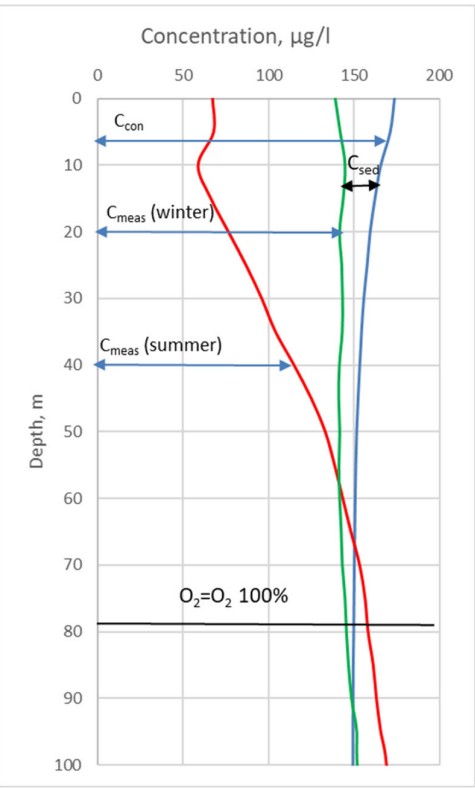

**Figure 2.** An example of the vertical distribution of nitrogen–nitrate concentrations in one of the regions of the Barents Sea. The red line is the winter distribution of nitrogen–nitrate $C_{meas}$ (winter); the green line is the summer distribution $C_{meas}$ (summer); the blue line is the summer conservative concentration; the black line is the horizon at which the saturation of water with dissolved oxygen in the summer decreases to less than 100%.

The measured concentration of a nutrient at a point will consist of conservative and non-conservative concentrations (1).

$$C_{meas} = C_{con} \pm \Delta C_{non-con} \tag{1}$$

$C_{meas}$—resulting (measured) concentration;

$C_{con}$—a conservative concentration that depends only on the mixing of different waters and is independent of photosynthetic or geochemical activity (sediment removal);

$\Delta C_{non-con}$—non-conservative part. This component consists of the "productive" component $C_{phyto}$ and the "non-productive" component $C_{sed}$;

$C_{phyto}$—change in concentration as a result of the photosynthesis process or mineralization of organic matter—the "productive" component;

$C_{sed}$—change in concentration as a result of an exchange with the underlying layers (precipitation of a given substance or its entry from the lower layer) as a result of vertical "non-productive" component mixing. This component also includes the inflow as a result of advection.

$$\Delta C_{non-con} = C_{phyto} + C_{sed} \tag{2}$$

To solve the problem of primary production determination by the consumption of a nutrients, it is necessary to determine the amount of change in the nutrients concentration at a point as a result of the photosynthesis process. To solve this problem, it is necessary to determine all of the biogenic element balance components.

2.1.1. Determination of the Conservative Component

The conservative concentration can be determined using the following equation:

$$C_{con} = f_a \times C_a + f_r \times C_r + f_i \times C_i \tag{3}$$

$$f_a + f_r + f_i = 1 \tag{4}$$

where:

$C_a$—the average concentration of the studied element in "purely" Atlantic waters;

$f_a$—the relative volume of "purely" Atlantic waters in the resulting water mass (%);

$C_r$—the average concentration of the studied element in "pure" river waters;

$f_r$—the relative volume of "pure" river water in (%);

$C_i$—the average concentration of the studied element in "ice" waters;

$f_i$—the volume of sea ice formed or melted, reduced to the density of water or the volume of meltwater in %, for simplicity, we will call this value "ice" waters.

The concentrations of the biogenic element $j$ ($Ca_j$, $Cr_j$, $Ci_j$) in the initial water masses are presented in Table 1.

**Table 1.** Values of the studied parameters ($Ca_j$, $Cr_j$, $Ci_j$) in the initial water masses.

| Base Water | P-PO$_4$ (µg/L) | N-NO$_4$ (µg/L) | Si-SiO$_3$ (µg/L) |
|---|---|---|---|
| Atlantic | 23.4 | 114.2 | 135.3 |
| River | 26.2 | 80.6 | 3061 |
| Ice | 11.1 | 24.8 | 30.9 |

The $Ca_j$ to the Atlantic waters in Table 1 refer to the average values obtained for the presented parameters of the Atlantic waters entering the Barents Sea for the layer 0–200 m for February, in the area limited by coordinates 71.5–73.5° N and 20–25° E. Values are selected from the general NODC database [22,23]. The time period of the year and the layer for the sample was chosen in order to ensure the maximum values of the average concentrations of nutrients, the decrease of which will begin in the spring, with the beginning of the photosynthesis process intensification.

The $Cr_j$ to the river waters in Table 1 refer to the average values obtained for 10 years (2006–2015) of the studied parameters' average annual concentrations for the river basins of the White and Barents Seas according to the yearbooks of surface water quality [24].

The $Ci_j$ to the "ice" waters in Table 1 refer to concentrations of nutrients obtained from materials published in the book Sea of the USSR, the Barents Sea [25], which were obtained in scientific research on the icebreaker "Otto Schmidt" in the 1980s of the 20th century.

The methodology for calculating $f_a$, $f_r$ and $f_i$ is presented below in the hydrological block (Section 2.2).

2.1.2. Determining the Values of the Non-Conservative Component ($\Delta C_{non-con}$)

The difference between the measured value and the conservative value will determine the value of "non-conservative".

$$\Delta C_{non-con j} = C_{meas j} - C_{con j} \tag{5}$$

$j$—the considered parameter (mineral phosphorus, silicon, nitrogen, or any other).

Then, at $\Delta C_j < 0$—there is a decrease in the concentration of a particular biogenic element due to non-conservative factors (in what follows we will not use the index (non-con).

Using Equations (1)–(5), you can calculate the total non-conservative concentration, which is the sum of both production and non-production components. Since the primary

production is determined for the volume of a unit area, the obtained values must be integrated from the surface to the horizon, where the measured concentration is equal to the winter concentration. Let us take as the lower limit of integration to the maximum horizon at which, during the year, the saturation of water with dissolved oxygen passes from values of more than 100% to values of less than 100%. In this case, the integration boundary can be any, provided that it is not higher than the maximum horizon at which this transition is observed, otherwise, the obtained values will be underestimated. However, in order to avoid the influence of other processes, it is better than the lower limit of integration to be closer to the above-indicated point. For each calculated point, the position of the transition horizon for the value of water saturation with dissolved oxygen was determined above 100% (*h*). For different areas of the sea, this value varied from 40 (or from the bottom layer) to 100 m.

For each month, the average monthly integrals value of $C_{con}$ and $C_{meas}$ from the surface to the selected horizon *h* are calculated, with the *dh* values presented above.

$$Q_m^n = \int_0^h C_{meas}^n \, dh \ \text{ and } \ Q_c^n = \int_0^h C_{con}^n \, dh \tag{6}$$

Index *n* indicates the month (1, 2, … 12).

For each month, the nonconservative component $\Delta Q^n$ was calculated as the difference between the values of the integrals $Q_m^n$ and $Q_c^n$.

$$\Delta Q^n = Q_c^n - Q_m^n \tag{7}$$

This value characterizes the total value of "non-conservative"—i.e., the total change in the biogenic element due to all processes of photosynthesis and mineralization of organic matter (productive component) and the processes of exchange of biogenic matter both vertically and horizontally (non-productive component). The plus sign (+) indicates the total amount of expenditure, the minus sign (−) indicates the total amount of income. If the change in the studied biogenic element depended only on the photosynthetic activity course and the mineralization of the formed organic matter, then this value, before the start of the photosynthesis process, would be equal to zero. If it is not equal to zero, then this means the inflow or outflow of this element is due to other processes, such as advection and exchange with underlying layers. The minimum value of the $\Delta Q^n$ for the year, which, before the start of the photosynthesis process (usually for different areas of the sea, this is February–March) will be the value of the non-productive component.

$$\Delta Q_{sed} = min \, (\Delta Q^n) \tag{8}$$

In this paper, we will assume that this value is constant throughout the year, but varies from region to region. If this value is positive, then there is an outflow of the biogenic element from the euphotic layer and vice versa. In this case, the balance of the biogenic element at the point can be written as:

$$Q_c^n - Q_{phyto}^n - \Delta Q_{sed} - Q_m^n = 0 \tag{9}$$

The component, which depends only on the photosynthetic activity and mineralization of the formed organic matter, is calculated for each month by the Equation (9).

$$Q_{phyto}^n = Q_c^n - Q_m^n - \Delta Q_{sed} \tag{10}$$

Equation (10) describes the change in the supply of a nutrients in the euphotic layer from month to month (*n* = 1,2,3 … 12). However, as shown in some works [2,16,17,20], this value decreases due to the return of this element to the marine environment as a result of its regeneration. In this case, taking into account the regeneration of the biogenic element, Equation (10) can be written in the following form:

$$Q_{phyto}^n = Q_c^n - (Q_m^n - Q_r^n) - \Delta Q_{sed} \tag{11}$$

where $Q_r^n$—is an increase in the reserve of a nutrients due to its remineralization.

In the primary production estimations by Arzhanova N.V., for the Sea of Okhotsk, the value of silicon regeneration was obtained to be 50% of the value already consumed by phytoplankton. In this work, the amount of regeneration was estimated for each region separately. For this, the rates of regeneration of nutrients were estimated by the rate of replenishment of their stock, when photosynthesis either stopped or was in an extremely slow state, and the replenishment of the nutrient stock from the underlying layers is hindered by the vertical water density gradient that has not yet been destroyed. As will be shown below, this phenomena usually occurs in October or November. In Equation (12), $(Q_{phyto}^{n-1})$ is the value from which the amount of nutrient regeneration for the month $n$ is determined.

$$Kj = \frac{\left[(QP_{phyto}^{n-1}) - (Q_{phyto}^{n})\right]}{(Q_{phyto}^{n-1})}, \tag{12}$$

The coefficient $Kj$ for the area in question is determined using the descending part of the curve (Figure 3), where the decrease of $Q_{phyto}^{n}$ from month $n-1$ to month $n$ is the largest in the year. The regeneration coefficient is assumed to be constant in a given area for the whole year, but the absolute value of the regeneration value will change from month to month since it is counted from the amount of the nutrients already consumed by that time. Of course, this value will depend on the water temperature and the species composition of phytoplankton, but the study and application of these dependencies will be considered at the next stage of improving this method.

As a result, the final value of nutrient consumption in month $n$, in the process of photosynthesis, will be determined by the Equation (13)

$$Q_{phyto}^{n} = Q_{c}^{n} - Q_{m}^{n} - \Delta Q_{sed} + (Q_{phyto}^{n-1}) \times Kj \tag{13}$$

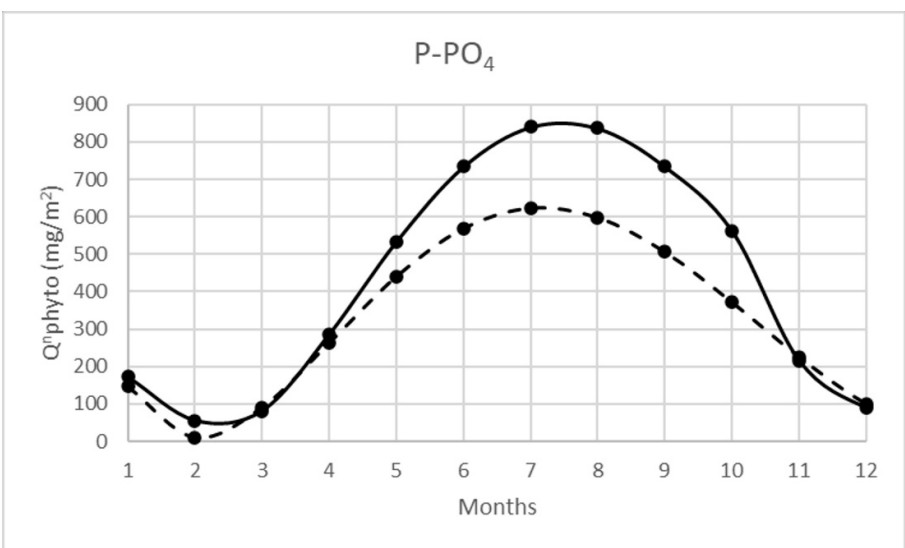

**Figure 3.** Example results assessment of the annual cycle nutrient consumption. Dotted line, excluding remineralization (Equation (11)), solid—taking into account remineralization (Equations (12) and (13)). In this figure, $n$ can take values on the descending branch of the graph (usually 10 or 11). In this case $n = 11$ is November, then $n - 1$—is October. ($Kj$ = (380 − 220)/380 = 0.42).

2.1.3. Calculation of Primary Production

For the transition from the value of nutrient consumption to primary production, i.e., to carbon consumption, at present, in most cases, the average stoichiometric Redfield–Richards ratios are used, which in the molar form are C:Si:N:P = 106:23:16:1 [1,6,13,14,26]. However, the application of these ratios for phytoplankton with different secies compositions can lead to large errors. Thus, the ratios of the presented elements differ significantly

in different dominant divisions of plankton. For example, the change in C:P in the phytoplankton from diatom to peridinium differs by 37%, the change in C:N by 24%, and C:Si by 14 times (Table 2). Accounting for the C:Si ratio variations, change is especially important, since, as mentioned above, the calculation of production for silicon reflects the total production (NCP$_{Si}$).

**Table 2.** Relative chemical composition of phytoplankton systematic groups by mass [13,26].

| Plankton | C (by Mass) | Si (by Mass) | N (by Mass) | Ph (by Mass) (Phosphorus) |
|---|---|---|---|---|
| Diatom | 100 | 93.0 | 18.2 | 2.7 |
| Peridinium | 100 | 6.6 | 13.8 | 1.7 |

Note: units of mass (μg, mg, g …).

In this work, an attempt was made to use individual stoichiometric ratios. In this case, it is assumed that all phytoplankton consists of only two systematic groups of phytoplankton—diatoms and dinoflagellates. This assumption also introduces an error in the calculations. However, if we evaluate the entire Barents Sea, then the average sum of the biomasses of these two phytoplankton systematic groups, according to the Biological Atlas of the Barents Sea [27], when analyzing 1000 samples, is 94%. That is, only 6% is the biomass of other systematic groups of phytoplankton. According to the results of some expeditionary field studies, the total biomass of these two groups in the Barents Sea was generally at the level of 99% [28].

The observed measured (real) stoichiometric ratio of carbon to one of the nutrients will be equal to the ratio of the carbon amount to the amount of silicon, nitrogen, or phosphorus (index *J*) involved in the process of photosynthesis:

$$w_J = \frac{(Q^n_{phyto})_C}{(Q^n_{phyto})_J} \tag{14}$$

The numerator is the amount of carbon involved in the process of photosynthesis, and the denominator is the amount of the biogenic element *J* involved in the process of photosynthesis.

If we assume that all phytoplankton consists of two systematic groups of phytoplankton—diatoms and peridiniums, then we can calculate the value of *wJ* using the values of the phytoplankton relative chemical composition according to X. Sverdrupat [13]) represented in Table 2.

$$dw_d + pw_p = w_j \tag{15}$$

where:

$d$—the relative contribution of carbon in diatom plankton biomass to total primary production;

$p$—the relative contribution of carbon in peridinium plankton biomass to total primary production;

$w_d$ and $w_p$—stoichiometric ratios of carbon to the *j*-th biogenic element for the systematic groups of diatoms and peridinium phytoplankton, respectively, calculated from Table 2.

Since we made the assumption that we have only 2 systematic groups of phytoplankton, then:

$$d + p = 1 \tag{16}$$

*wJ*—stoichiometric ratios which are calculated from the amount of consumption of nutrients in the process of photosynthesis. Index *j*—Si, N, or Ph. In this work, for further calculations, we used the ratios of nutrients Si/Ph and Si/N.

$$w_{Si/Ph} = \frac{(Q_{phyto}^n)_{Si}}{(Q_{phyto}^n)_{Ph}} \tag{17}$$

Using Equations (14)–(17), it is easy to calculate the values of $d$ and $p$.

$$d = \frac{w_{Si/Ph} - w_P}{(w_d - w_p)} \tag{18}$$

In this case, $w_d$ and $w_p$—are the $Si/Ph$ ratio for diatom and peridinium plankton calculated from Table 2. Value $k_{Si/Ph}$—calculated by Equation (17). Equation (19) describes the calculation of the current or individual stoichiometric ratio of carbon to nutrient after calculating the values of $d$ and $p$.

$$dw_d + pw_p = \frac{(Q_{phyto}^n)_C}{(Q_{phyto}^n)_J} \tag{19}$$

As a result, using Equation (19) and knowing the values of $d$ and $p$, the primary production is calculated.

$$(Q_{phyto}^n)_C = (Q_{phyto}^n)_j \times (dw_d + pw_p) \tag{20}$$

In Equations (19) and (20) index $C$ is carbon consumption (primary production). In this case, this is the value ($NCP_{Si}$), if the calculation for silicon was used ($j = Si$, $w_d$ and $w_p$—are the stoichiometric ratios of carbon to silicon calculated from Table 2) or the value ($NCP_N$) if the calculation for nitrogen was used ($j = N$, $w_d$ and $w_p$ are stoichiometric ratios of carbon to nitrogen calculated according to Table 2).

### 2.2. Methodology for Calculating the Values of $f_a$, $f_r$ and $f_i$ (Hydrological Block)

To calculate the conservative concentration (Equation (3)), it is necessary to know the content of basic waters (the Atlantic, river, and ice) in the sample under study. The materials of the public databases NODC [22] and NASA [10] were used in this work.

The calculation of $f_a$, $f_r$, and $f_i$ is carried out according to salinity and isotopic parameter, taking into account corrections for the predominance of the ice formation process over the ice melting process ($f_i < 0$). To estimate the values of the relative volumes of the Atlantic, river, and ice waters, a three-component system of mixing equations was used in various works [18,27–47] and others:

$$fa \times Sa + fr \times Sr + fi \times Si = Smeas$$
$$fa \times Ia + fr \times Ir + fi \times Ii = Imeas \tag{21}$$
$$fa + fr + fi = 1$$

where:

$S_r$—salinity of "pure" river waters, which is always zero ($S_r = 0$) (psu);
$S_i$—sea ice salinity (psu);
$S_a$—salinity of "pure" Atlantic waters (psu);
$I_a$—the value of the isotope parameter $\delta^{18}O$ for "purely" Atlantic waters (‰);
$I_r$—for "pure" river waters (‰);
$I_i$—for ice waters (‰);
$f_a$, $f_r$ and $f_i$—content of Atlantic, river, and ice waters (%);
$S_{meas}$—resulting (measured) water salinity (psu);
$I_{meas}$—resulting (measured) value of the isotope parameter $\delta^{18}O$ (‰).

This system of equations can be used only in cases where the ice-melting process predominates ($fi > 0$) [21]. In the case when the ice formation process prevails ($fi < 0$), the system of Equation (21) is supplemented by two equations [26]:

$$fr^w = fr - |fi| + k \ x \ |fi|/(1 + k) \tag{22}$$

$$fr^w = fr - |fi| + k \; x \; |fi|/(1 + k)$$

$$fa/fr = k$$

(23)

For the first time, the not entirely legitimate use of the system of Equation (21) for the case of the predominance of ice formation processes was shown in the work of Dubinina E.O. et al. [37], but this paper shows the calculation method only for river waters.

In this case, $fa^w$ and $fr^w$ are the content of Atlantic and river waters in the subglacial water layer [34]; $fa$ and $fr$—values of the content of Atlantic and river waters calculated by the system of Equation (21). The final values of salinity and isotope parameter for calculations are presented in Table 3.

**Table 3.** End-member values used in mass balance calculations.

| Sa (psu) | [3] Si (psu) | Ia (‰) | Ir (‰) | Ii (‰) |
|---|---|---|---|---|
| 35.14 [1] | 5.86 [2] | 0.35 [3] | −14.23 [4] | [5] Values in the surface + 1.96‰ |

[1] Accepted by the author on the database NODC [22,23] for the waters of the Barents Sea, as the median value for square 10–17 E, 71–75 N, for 150–250 m, for February; [2] Calculated by 288 definition of the salinity of ice presented in the book of the Sea of the USSR. Barents Sea [25]; [3] Accepted by the author on the database [22] for this square (p. 1); [4] Accepted as a weighted average based on on the Kola, Severnaya Dvina, Pechora and Pinega rivers, including annual flow [48]; [5] Jointly determining the value $\delta^{18}O$ of sea water under the ice and in the ice in the Barents Sea (6 samples 1.81 ± 0.34 see addition information) and combined with data Melling and R. M. Moore [27] (7 samples 2.09 ± 0.38) in the Beaufort Sea.

In this work, the results of determining $\delta^{18}O$ and salinity, which are available in public databases, were used. These data were collected between 1972 and 2008 and published on the NASA [10] and NODC [22] websites. The list of data used is presented in [45]. In addition, the results of studies in 2014 were used [41]. The error in measuring $\delta^{18}O$ was 0.03–0.07‰ for studies conducted in 1993–1995, and 0.1–0.2 $\delta^{18}O$‰ for studies conducted before 1989 [33]. In addition, this work used data from the parallel determination of $\delta^{18}O$ n ice and in the under-ice water layer as part of the expedition aboard the R/V Dalnie Zelentsy in 2021 in the Barents Sea, as well as generalized data on the determination of the $\delta^{18}O$ isotope parameter in the rivers of the Barents and White Seas (with an error ±0.025‰) presented in Table 3.

Since we are using nutrients and salinity data at the grid points, we need to know the value of the isotope parameter for each salinity value. However, there is no such data. Many studies have shown that the dependences between salinity and $\delta^{18}O$ have a correlation coefficient close to 1. The generalized dependence obtained from the results of 2182 measurements at 319 stations in the Barents Sea, based on the results of studies by different groups of authors, is presented in [45] (Figure 4).

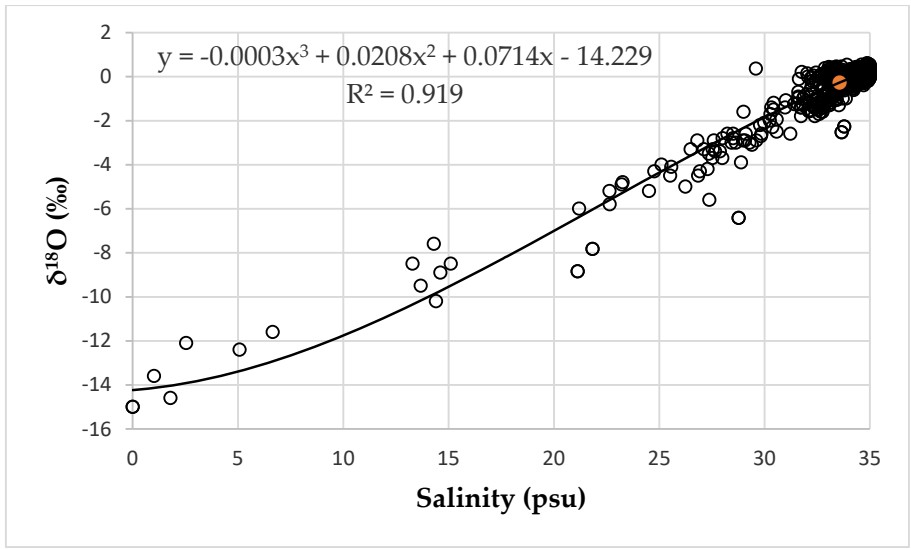

**Figure 4.** Relationship equation between salinity (psu) and δ¹⁸O (‰) in the Barents Sea.

In this case, the correlation coefficient is close to 1 ($R^2 = 0.91$). Since this graph represents all data, regardless of the season and sampling horizon, we assume that the resulting relationship equation is applicable for all salinities, regardless of whether the δ¹⁸O value was measured along with salinity or not. The proposed hypothesis was tested by dividing the series into two series, in fact, randomly, with 1100 values in each. The equation for the relationship between salinity and δ¹⁸O was calculated based on the data from the first series. Then, using this equation for the second series, δ¹⁸O values were calculated from the salinity values. Further, the calculated and measured values of δ¹⁸O were compared with each other. The significance of linear regression between measured and calculated δ¹⁸O values was tested using Fisher's f-test. Student's *t*-test was used to check the equality of the mean values in two samples. For both tests, the significance level was less than 0.05 (Table A1). Therefore, our assumption is valid and for each salinity value, it is possible to calculate the value of δ¹⁸O even outside the series of mutual determination of these parameters. In this work, the values of the isotopic parameter δ¹⁸O for each salinity value were calculated using the equation presented in Figure 4.

*2.3. Conclusion to the Section*

The collection of all data, as well as the calculation of integrals (Equation (6)), was carried out using the Ocean Data View software package [49].

Statistical processing, including the calculation of linear trends, testing the significance of linear regression (Fisher's f-test), as well as checking the equality of the means in two samples (Student's *t*-test) were carried out using the EXCEL software package.

In total, calculations for the Barents Sea were made for values at ~550 grid points (Figure 1). All calculations for Equations (1)–(7), including Equations (21)–(23), as well as the regression equation in Figure 4, were carried out for each month at each calculated point and at each horizon (presented above, from 5 to 25 horizons). Further, the integration results obtained by Formulas 6 and 7 were averaged for regions 10° in longitude (from 0° E) and 2° in latitude (from 68° N). In total, there were 35 squares. Further calculations were carried out on average for each square.

When developing this method, three assumptions were made:

(1) The contribution of the non-productive component ($\Delta Q_{sed}$) varies from region to region but is constant throughout the year in the region.

(2) The entire phytoplankton of the Barents Sea consists of only two systematic groups of phytoplankton—diatoms and dinoflagellates (peridinium). If we evaluate the entire Barents Sea, then the average sum of the biomasses of these two phytoplankton systematic groups, according to the Biological Atlas of the Barents Sea, when analyzing 1000 samples, is 94%.

(3) The regeneration coefficient (Equation (12)) varies from region to region but is constant throughout the year in the region. The absolute value of the regeneration value will vary from month to month.

The aim of the present work is first to show a new application of water stable isotope use, in particular, the parameter δ¹⁸O, in assessing the change in nutrient concentrations and calculating primary production. In the future, with the improvement of this technique, the accepted assumptions will be specified, and the resolution of calculations will also increase.

## 3. Results

### 3.1. Verification of the Results Obtained

Before analyzing the obtained materials, it is necessary to verify the obtained materials with the results worked out by other methods. It makes no sense to compare the absolute values of primary production and measured values of phytoplankton biomass. Nevertheless, it is possible to compare the relative values of diatoms and dinoflagellates' contribution to the total production (calculated values), on the one hand, and to the total biomass in the case of its direct measurement (Figure 5), on the other hand.

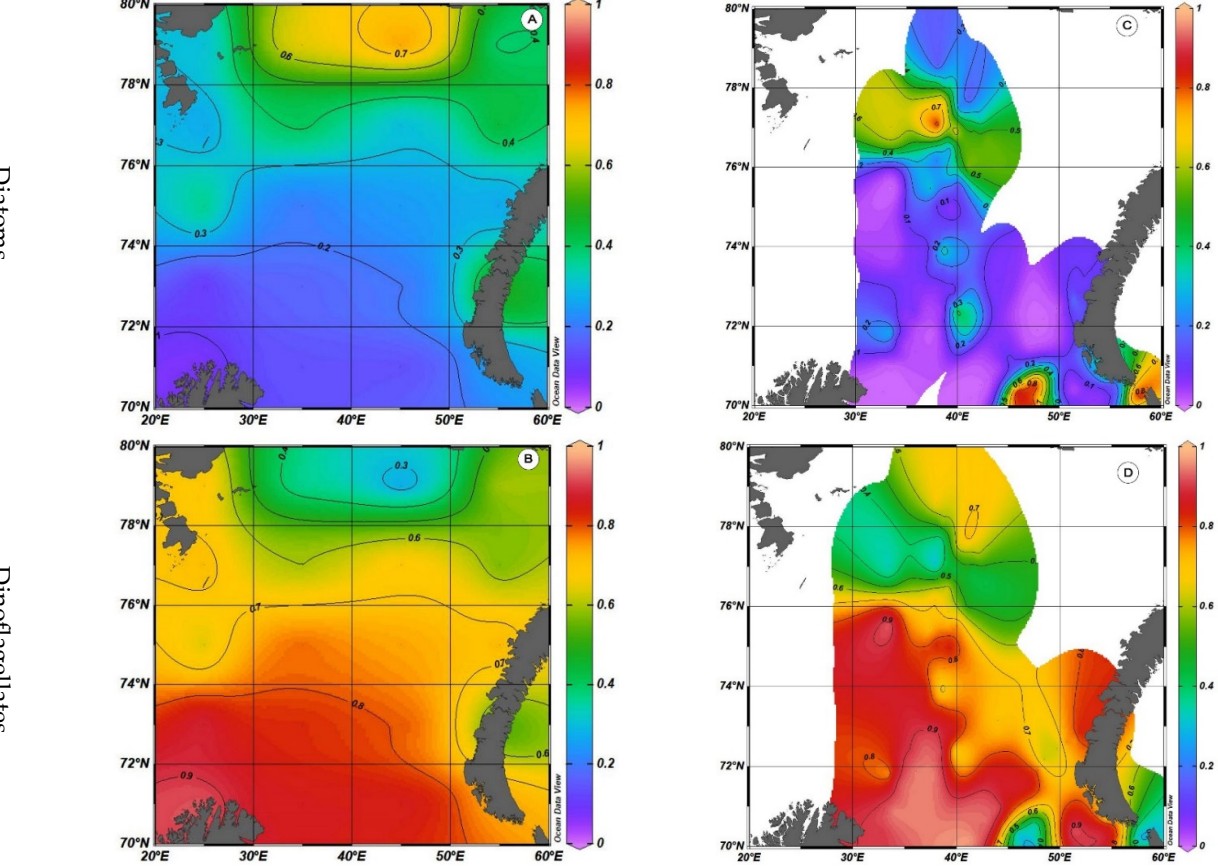

**Figure 5.** Relative values of total sum diatoms "d" (**A**) and dinoflagellates "p" (**B**) contribution to the total production (calculated values Equation (16)). Relative contribution diatoms (**C**) and dinoflagellates (**D**) to the total phytoplankton biomass (direct measurements), July—September.

The biomass values are from the archive of the Murmansk Marine Biological Institute, Russian Academy of Sciences, from 1970 to 2013, for the season from July to September, the period with the most data available. In addition, this comparison may not seem entirely correct, since for the case of calculated values, integrated values are presented for the depth of the euphotic layer, and for the case of direct measurements of biomass, only the surface layer is presented. Nevertheless, the nature of the distribution of these values, as well as the values of the relative contribution values of these two systematic groups, are quite close. In addition, by the beginning of summer, the intensity of the photosynthesis process reaches its maximum and reaches a plateau, which leads to the maximum values of primary production—and the maximum values of the decrease in the nutrient supply. Under these conditions, the relative values of calculation error are minimal.

The results for the verification of the calculated values of the diatom and dinoflagellate systematic groups' relative contribution to the total primary production with direct measurements of these systematic groups' contribution to the total phytoplankton biomass showed the similarity of not only the nature of the distribution but also the values themselves.

The results of our calculation in the Barents Sea can be compared to Reigstad et al. [6] for the water area, surrounding the Svalbard archipelago. In more detail, the results of this comparison are presented in the work of Namyatov et al. [48]. In work [6], calculations of primary production are made based on hydrodynamic and ecosystem models. The hydrodynamic model is a large-scale model with a horizontal grid spacing of 20 km and a nested model resolution of 4 km. The ecosystem module includes nitrates, ammonium, silicates, diatoms, flagellates, microzooplankton, bacteria, heterotrophic nanoflagellates, fast-sinking detritus, slow-sinking detritus, and two groups of mesozooplankton *Calanus finmarchicus* and *Calanus glacialis.*

Based on the modeling results, water areas with different PP values were identified in this area. Because the resolution of the used model was much higher than in our investigations, as a result, six regions were identified (Table 4). According to the results of our study, there are three identified regions, but we believe that the boundaries of the three regions that we obtained include the water areas of six regions from the article above [6] (Table 4), and the results have pretty good convergence. A particularly high degree of similarity was obtained from the PP estimate in Region III, adjacent to the archipelago in the northeastern part and extending further to the northeast of the sea. According to the results of both studies, a decrease in PP values is two times compared to other areas. However, according to available data [40], the relative value of the standard deviation from the mean value of PP in this area is several times greater than that presented in the compared work.

**Table 4.** Average primary production rates and standard deviations in regions of the Svalbard) (accumulated primary production in August).

| Ref. [6] | | * This Study and [48] | | |
|---|---|---|---|---|
| Region | Average NCP$_{Si}$ gC/m$^2$ | Average NCP$_{Si}$ gC/m$^2$ | Max NCP$_{Si}$ gC/m$^2$ | Region |
| Svalbard waters province | $100 \pm 7$ | $90 \pm 34$ | 140 | |
| Western Svalbard Shelf (SS) | $106 \pm 8$ | | | |
| West Spitsbergen Current (WSC) | $134 \pm 8$ | $86 \pm 7$ | 100 | II |
| Fram Strait East | $108 \pm 6$ | | | |
| Svalbard Bank Frontal System | $112 \pm 8$ | $117 \pm 15$ | 140 | I |
| Hindlopen Strait | $67 \pm 12$ | $56 \pm 48$ | 103 | III |
| Svalbard North | $54 \pm 10$ | | | |

Note: * Regions I, II, and III are the waters surrounding the Svalbard archipelago with a relatively uniform value of NCP$_{Si}$.

### 3.2. Patterns of Primary Production and Nutrient Consumption Distribution

This technique makes it possible to estimate the annual cycle of each nutrient's consumption, as well as to identify the limiting element of the photosynthesis process in a particular area.

The results of each nutrient consumption values are calculated according to (Equation (10)) from the stock value ($\Sigma\Delta P_{reserv}^n$), taking into account the inflow/outflow of this element due to the non-productive component ($\Delta P_{sed}$).

$$\Sigma\Delta P_{reserv}^n = \Delta P_c^n + \Delta P_{sed} \tag{24}$$

The distribution of this value for the period of the photosynthesis process maximum development from 1 July to 30 September is shown in Figure 6.

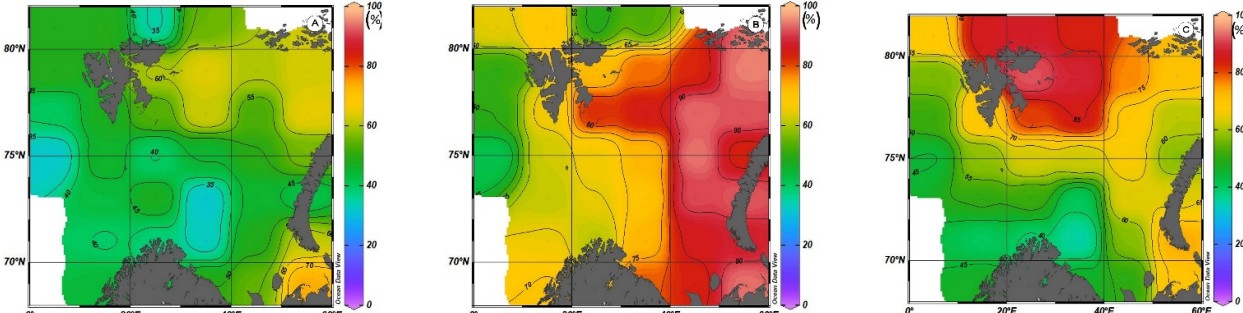

**Figure 6.** Nutrients (P-PO$_4$ (%) (**A**) N-NO$_3$ (%) (**B**) and Si-SiO$_3$ (%) (**C**)) intake average values for the period from 1 July to 30 September ($Q_{phyto}^n/(Q_c^n \pm \Delta Q_{sed})$ × 100).

The results of these evaluations show that:

1. Throughout the Barents Sea, phosphate phosphorus is not a limiting factor in the development of the photosynthesis process. The consumption of this element increases from west to east from 40 to 70%.
2. The value of the nitrogen–nitrate total consumption actually in the entire water area of the eastern part of the sea in the summer, from the coast to the Franz Josef Land archipelago, is at the level of the full consumption of this element in the process of photosynthesis and is 90–95%. In the western part of the sea, in waters that are composed of the maximum incoming Atlantic waters, the values of nitrogen consumption are much lower and are in the range from 40 to 70%, although, as will be shown below, the values of primary production in these waters are maximum.
3. The distribution of silicon consumption in the southern and central parts of the sea also tends to increase from west to east in the range of 30 to 60%. However, in the northern part of the sea, between the Svalbard and Franz Josef Land archipelagos, silicon can be a limiting factor in the development of the photosynthesis process.

Based on the results of calculations, maps of the average monthly concentrations of primary production distribution in the Barents Sea by silicon and nitrogen were prepared. Figure 7 shows the distribution of NCP$_{Si}$ (Figure 7A) and NCP$_N$ (Figure 7B) values. According to the distribution of primary production, six centers can formally be distinguished in the Barents Sea, in four of which, there are increased values of this value (I–IV4), in two—lower values (V–VI).

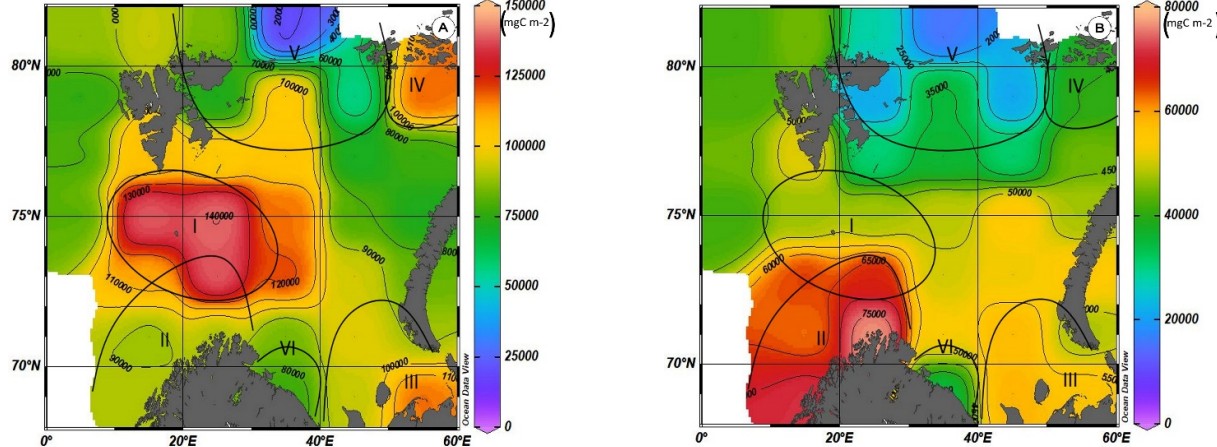

**Figure 7.** Estimated values of primary production (mg C m⁻²) for silicon (**A**) and nitrogen (**B**). Average values for the period from July to September. The numbers I to VI indicate the centers of regions with maximum or minimum values of primary production.

### 3.2.1. Nutrient Consumption and Primary Production in the Area with the Highest PP Values

The maximum values of primary production are observed in *Region I*, adjacent on all sides to the Medvezhinsky shallow water (Figure 7). This area is located in the zone of inflow of Atlantic waters. The average values of total primary production for silicon ($NCP_{Si}$) in this water area during the period of maximum development of photosynthesis (August–September) are at the level of ~160 g C m⁻² (152 ± 36 g C m⁻²), with maxima up to ~190 g C m⁻². Increases in the values of primary production on average in this area are observed until November (~100–120 g C m⁻²), and then there is a sharp decline by December to 56 ± 14 g C m⁻². The average values of total primary production for nitrogen $NCP_N$, during the period of maximum development of photosynthesis in this region, are 100 g C m⁻² less than $NCP_{Si}$ and amount to 58 ± 9 g C m⁻² (Figure 8).

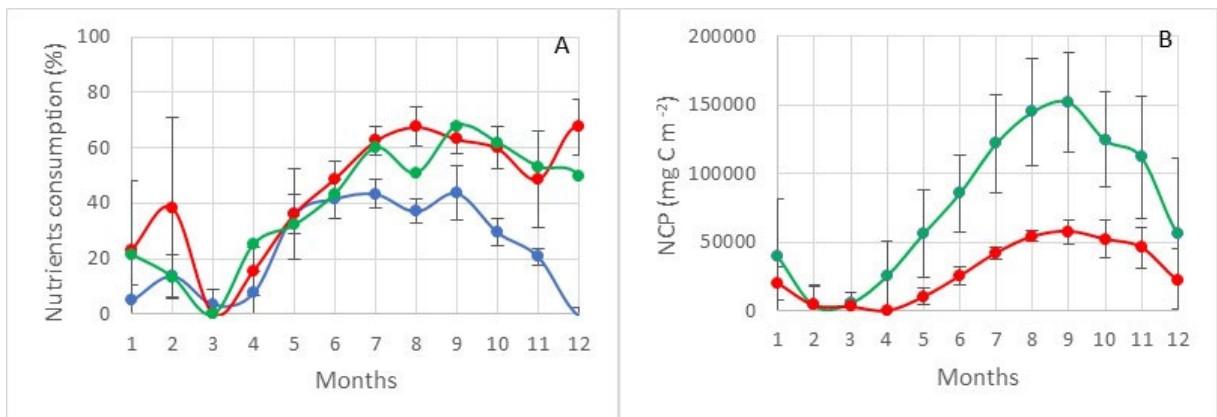

**Figure 8.** Annual cycle of primary production and nutrient consumption in the Region I: (**A**) consumed part of the nutrient supply in the process of photosynthesis (blue: mineral phosphorus, red: mineral nitrogen, green: silicon). (**B**) primary production (green: $NCP_{Si}$; red: $NCP_N$).

The western border of the region is located on the slope of the depths. Depths in the water area vary here from 500 m or more in the west of the area to 25 m off the coast of Medvezhiy Island. The increase in PP values in this water area is associated with the entry of nutrients into the euphotic layer from the lower layers, more saturated with these sub-

stances. The same effect is also described in the Sea of Okhotsk in the area of the Kashevarov Bank [2], and the maximum values of PP are also observed here. However, in the Sea of Okhotsk, this effect is associated with the presence of a cyclonic eddy, resulting in the rise of water from the lower layers and the entry of nutrients into the euphotic layer. There is no mention in the literature about the presence or absence of such a cyclonic eddy in the Medvezhinsky–Spitsbergen shallow area, but when the Atlantic waters move from the west to the Barents Sea, it is obvious that they rise from depths where photosynthesis does not occur, and if south of shallow water, at depths of 250–300 m, not all water from deep layers, more saturated with nutrient layers, enters the euphotic layer, then in shallow water these waters directly enter the photosynthesis layer.

In October–December, if we take into account the concentration of dissolved oxygen, the intensity of the photosynthesis process is minimal, and the high values of nutrient consumption are supported by the residual values of non-mineralized organic matter. A sharp decrease in these values by December–January is caused by an increase in the intensity of vertical mixing processes and replenishment of the supply of nutrients from the bottom layers.

The structure of consumption of mineral forms of phosphorus, nitrogen, and silicon is shown in Figure 8. None of the nutrients considered is limiting to the process of photosynthesis. At the maximum development of the photosynthesis process, the residual reserves of silicon and nitrogen are at the level of 35–40%, and phosphorus at the level of ~60%, shown in Figure 8 (100% minus value of consumption).

### 3.2.2. Nutrient Consumption and Primary Production in Areas with High PP Values

Region II. This area is located in the southwestern part of the sea (Figure 7). Through its water area, there is a flow of Atlantic waters into the Barents Sea. In addition, this area is located in the zone of the continental slope on the western border of the Barents Sea shelf. The depth difference here ranges from 200–500 m in the eastern part of the region and up to 2000–3000 m in its western part. The depth of the lower boundary of the euphotic layer is 50–80 m. In contrast to the Region I, in Region II, the largest $NCP_{Si}$ values are 50 g C $m^{-2}$ less (103 ± 11 g C $m^{-2}$). In addition, in this region, the difference between the highest values of $NCP_{Si}$ and $NCP_N$ (73 ± 7 g C $m^{-2}$) is not so large and is only about 30 g C $m^{-2}$ (Figure 9B). In this area, none of the studied nutrients are the limiting factor of photosynthesis. The residual reserve of nitrogen during the period of maximum development of the photosynthesis process is at the level of 25–30%, and phosphorus and silicon are at the level of 55–60% as can be seen this in Figure 9A (100% minus value of consumption).

Apparently, in this area, there is an effect of the influence of the continental slope on the value of PP. A similar effect is given in works describing the enrichment of the euphotic layer with nutrients on the continental slope of Sakhalin and Kamchatka [2].

Region III. This area is located in the southeastern part of the Barents Sea, the waters of which are most affected by the flow of the Pechora River (Figure 7). The highest value of $NCP_{Si}$ in this area is 106 ± 7 g C $m^{-2}$ and for new production 65 ± 4 g C $m^{-2}$ (Figure 10B).

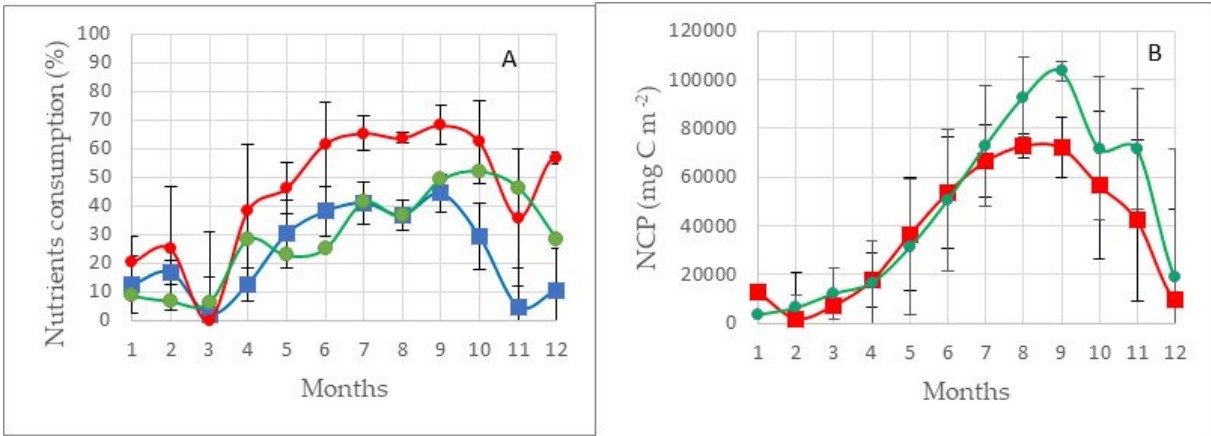

**Figure 9.** Annual cycle of primary production and nutrient consumption (**A**) and primary production (**B**) in Region II; (**A**) is the consumed part of the nutrient supply in the process of photosynthesis (blue: mineral phosphorus, red: mineral nitrogen, green: silicon); (**B**) primary production (green: NCP$_{Si}$; red: NCP$_N$).

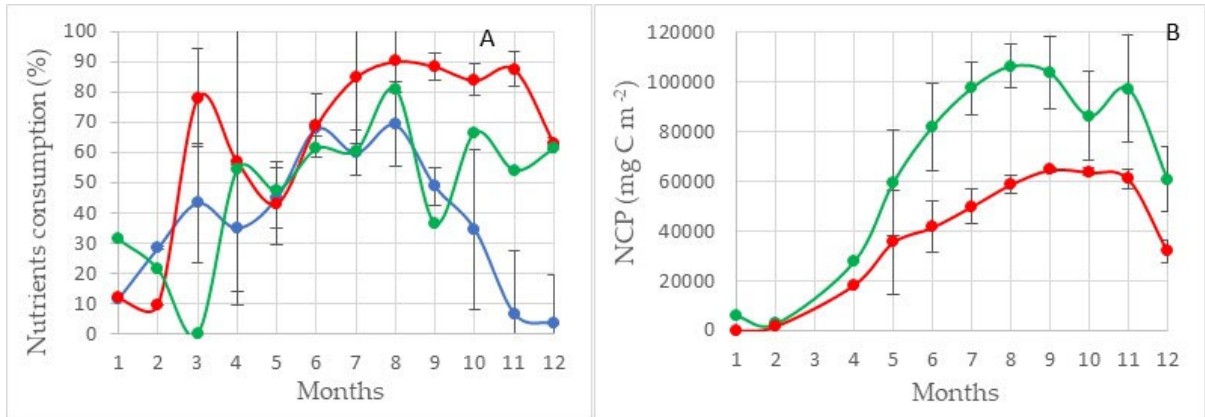

**Figure 10.** Annual cycle of primary production and nutrient consumption (**A**) and primary production (**B**) in Region III; (**A**) is the consumed part of the nutrient supply in the process of photosynthesis (blue: mineral phosphorus, red: mineral nitrogen, green: silicon); (**B**) primary production (green: NCP$_{Si}$; red: NCP$_N$).

In contrast to areas located in the zone of Atlantic waters, where the remaining stock of none of the studied nutrients reaches a critical value—limiting the course of photosynthesis, in this area, nitrogen can be such a limiting factor. During the period of maximum development of the photosynthesis process (August–September), the residual reserve of nitrogen is at the level of 10%, the largest value of its consumption is observed in August with amounts to 90 ± 6%. The level of silicon consumption this month is 80 ± 2%, and phosphorus is 70 ± 6%.

In the entire southeastern part of this area to Kolguev Island, the euphotic layer occupies the entire water column from the surface to the bottom, and in the deeper northwestern part, from 45 to 75 m. Another feature of this region, which affects the course of the photosynthesis process, is that, according to long-term average data, until May, almost the entire water area of this region is covered with ice (Figure A1). By May, the ice edge recedes to the east, but the entire southeastern part between the Novaya Zemlya archipelago and the coast of the mainland and from Kolguev Island to the Kara Gates is covered with ice [25]. Even in June, not all of the area is ice-free. As measured by phytoplankton studies in the area, even in the spread of light due to the ice cover, the start of the photosynthesis process in the spring does not stop. This effect was described in the works of

employees of the Murmansk Marine Biological Institute [50]. The presented calculations in Figure 10A,B show that already in April, in the presence of a continuous ice cover, the consumption of nutrients in the process of photosynthesis is noted, and the NCPSi values reach 25 g C m⁻². These results are confirmed by direct determinations of phytoplankton parameters [51]. The increase in PP is observed until August. The highest PP values in this region in August are 95 g C m⁻² for silicon and 39 g C m⁻² for nitrogen (Figure 10B).

The number of phytoplankton in this period is 150–550 thousand cells/L (which is approximately 450 µg/L) with a maximum in shallow water in the southern part. The maximum calculated PP values are also confined to the southern part of the region and amounted to 154 g C m⁻² for silicon and 66 g C m⁻² for nitrogen. According to the results of direct studies on phytoplankton parameters in October, it was shown that the total number of organisms decreased by 100 times and varied in the range of 0.5–3 thousand cells/L, and their biomass in the range of 2–70 µg/L [51]. Nevertheless, the calculated values of PP accumulation increased only by a factor of 5 and averaged about 20 g C m⁻² for silicon. Such a delay in the decrease in the calculated PP values relative to the decrease in the values of the total abundance and biomass of phytoplankton is explained by the rate of mineralization of organic matter accumulated in the euphotic layer, which is much less than the natural rate of decrease in the total abundance and biomass of phytoplankton in the autumn period.

Region IV. This area is influenced by cold currents directed from the Arctic Basin or the Kara Sea, as well as the coastal current of Franz Josef Land (Figure 7). According to long-term average data, seasonal ice processes occur throughout the year in the water area of this region (Figure A1). From April to June, the entire water area of this region is under ice. By July, the ice edge rises to the north, and half of the region's water area is free of ice. In August, according to long-term average data, most of the region is open from ice, but in the north, in the region of 79–80° N, the region's water area can be covered with ice. Since September, the reverse movement of the ice edge to the south begins.

The annual cycle of nutrient consumption and primary production is shown in Figure 11A,B respectively.

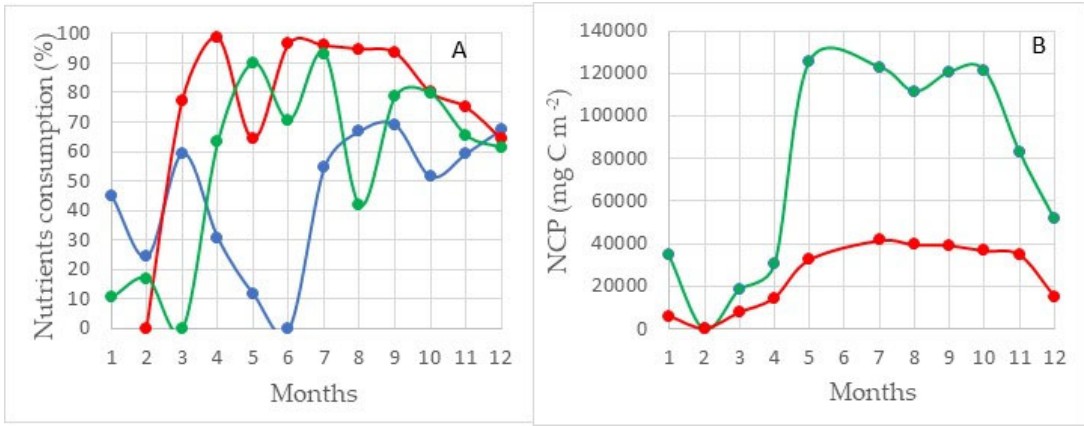

**Figure 11.** Annual cycle of primary production and nutrient consumption (**A**) and primary production (**B**) in Region IV; (**A**) is the consumed part of the nutrient supply in the process of photosynthesis (blue: mineral phosphorus, red: mineral nitrogen, green: silicon). (**B**) primary production (green: NCPSi; red: NCPN).

This region, along with the areas on the northern border of the sea, is the region with the least data, which is probably why the scatter of data in the annual cycle is very large. Nevertheless, it is possible to trace some patterns of changes in these parameters.

The level of nitrogen consumption in this area is a maximum and from April to August is in the range of 93–99% of the total supply of this element. The level of silicon consumption is also high and amounts to 75–95%, although the variability of this value is

quite large. The value of phosphorus consumption during the active phase of the photosynthesis process from April to September also has significant variability, but the highest consumption of mineral forms of phosphorus is 60–70% (Figure 11A). The highest $NCP_{Si}$ values are observed from May to October and are in the range of 110–125 g C m$^{-2}$. $NCP_N$ values are three times less and during this period are in the range of 35–40 g C m$^{-2}$ (Figure 11B).

In this area, the limitation of the process of photosynthesis by nitrogen consumption is clearly expressed, in fact, up to 100%.

3.2.3. Nutrient Consumption and Primary Production in Areas with the Lowest PP Values

Region V. This area is located on the northern border of the sea between the Spitsbergen archipelagos and Franz Josef Land and is characterized by the fact that from the north and northwest, it adjoins the continental slope zone of the Barents Sea shelf (Figure 7). A characteristic feature of this area is the inflow of Atlantic waters that passed through the Fram Strait in the western part of this area, as well as the inflow of these same waters from the north to the northern and eastern parts of this area [51]. Depths in this area vary from 100 m in the southern part to 500 m in the northern and northwest area. It should be noted that near the islands of the archipelago, the depths can decrease to 30–40 m. The lower boundary of the euphotic layer lies at a depth of 40–45 m. The characteristics of ice conditions are approximately the same as in Region IV. The difference is that, according to average long-term data, ice completely covers the water area of the region.

The lower boundary of the euphotic layer reaches depths of 50 m. The annual cycle of nutrient consumption and primary production is shown in Figure 12A,B.

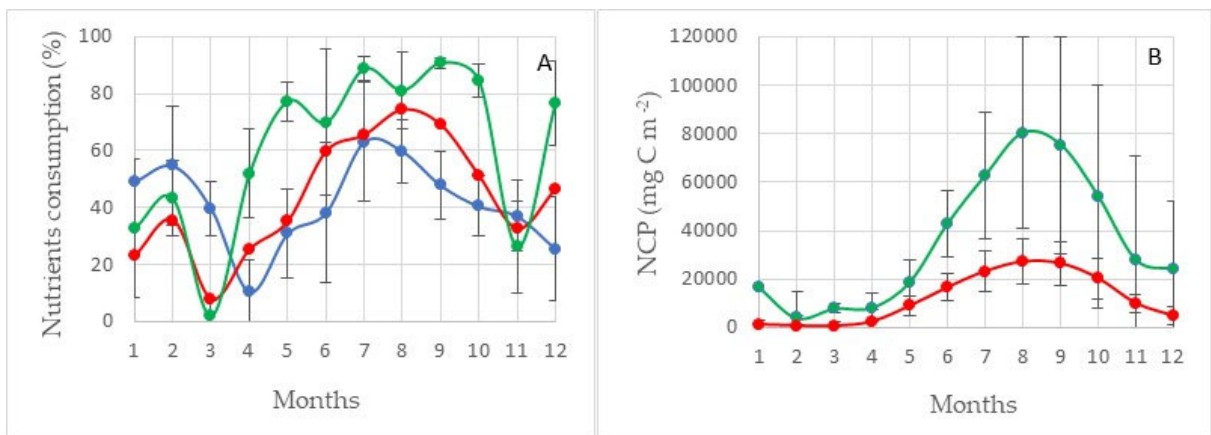

**Figure 12.** Annual cycle of primary production and nutrient consumption (**A**) and primary production (**B**) in Region V; (**A**) is the consumed part of the nutrient supply in the process of photosynthesis (blue: mineral phosphorus, red: mineral nitrogen, green: silicon); (**B**) primary production (green: $NCP_{Si}$; red: $NCP_N$).

In this region, in contrast to the regions discussed above, the limiting factor in the development of the photosynthesis process is not nitrogen, but silicon. The maximum consumption of silicon reaches 94%, the maximum consumption of nitrogen is at the level of 86%, and the average for the region does not exceed 80%. The maximum value of the average monthly values of $NCP_{Si}$ is 80 ± 45 g C m$^{-2}$, and $NCP_N$ is 27 ± 9 g C m$^{-2}$. Another feature of this area is the large values of the standard deviation of the calculated values of primary production. This feature is explained by the presence of several water masses, as well as the seasonal influence of ice phenomena.

Region VI. This area adjoins the coast of the Kola Peninsula and extends to the west from Varanger Fjord to the mouth of the White Sea. The waters of the western part of this region are formed by the waters of the Murmansk coastal current, which is part of the

Atlantic water flow. The lower boundary of the euphotic layer reaches depths of 75–80 m. The annual cycle of nutrients consumption and primary production is shown in Figure 13A,B.

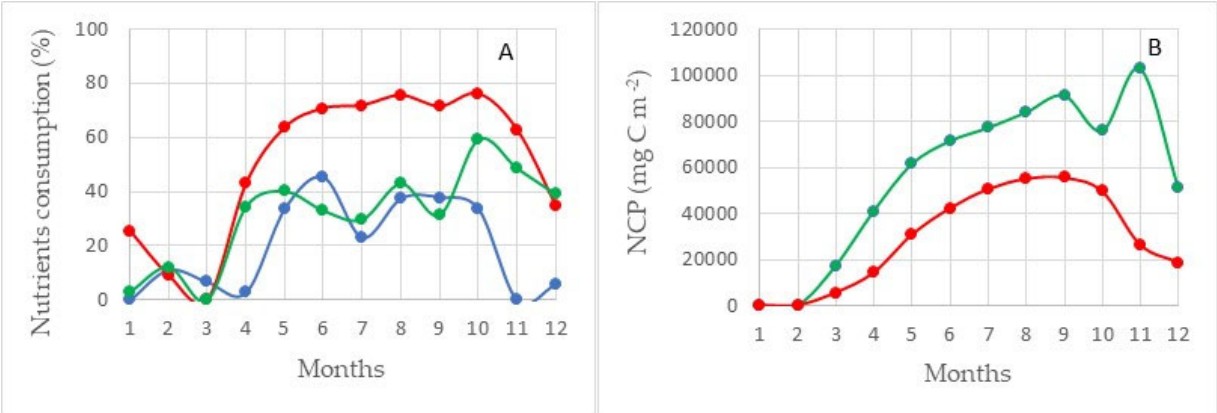

**Figure 13.** Annual cycle of primary production (**A**) and primary production (**B**) in Region IV; (**A**) is the consumed part of the nutrient supply in the process of photosynthesis (blue: mineral phosphorus, red; mineral nitrogen, green: silicon). (**B**) primary production (green: $NCP_{Si}$; red: $NCP_N$).

The annual cycle of consumption of nutrients and primary production corresponds to certain annual cycles in Region II. The main difference lies in the average values of new production, which in the early summer period is the period of maximum intensity of the photosynthesis process, by about 20 g C m$^{-2}$ lower than in Region II. A similar decrease in the calculated values of primary production to ~50-70 g C m$^{-2}$ is observed in the waters adjacent to the Kola Peninsula [6].

## 4. Discussion

Based on the proposed methodology, using the parameter of stable isotopes $\delta^{18}O$, calculated results of annual cycles of nutrient consumption and primary production in various areas of the Barents Sea, as well as the distribution of these values over the entire sea area, were obtained.

Verification of the results for direct phytoplankton measurements and data calculated by us showed high convergence. Verification of the results for the calculation of primary production with data presented by other methods and other teams of authors also gave good convergence values.

Some regularities in the dynamics of the consumption of nutrients and primary production both in the annual cycle and in the sea are revealed:

- The value of the total consumption of nitrogen–nitrate in the entire water area of the eastern part of the sea in the summer, is at the level of the total consumption of this element in the process of photosynthesis (90–95%). That is, the almost complete consumption of mineral nitrogen in this part of the sea is the limiting factor in the development of the photosynthesis process. In the western part of the sea, in waters that are composed of the maximum incoming Atlantic waters [52], the values of nitrogen consumption are much lower and are in the range from 40 to 70%, although the values of primary production in these waters are a maximum.
- The distribution of silicon consumption in the southern and central parts of the sea also tends to increase from west to east in the range of 30 to 60%. In the northern part of the sea, between the Svalbard and Franz Josef Land archipelagos, silicon can be a limiting factor in the development of the photosynthesis process.

- Throughout the Barents Sea, phosphate phosphorus is not a limiting factor in the development of the photosynthesis process. The consumption of this element increases from west to east from 40 to 70%.
- According to the largest values of primary production in the Barents Sea in the summer–early autumn period, an area with the highest average values of this value ($NCP_{Si} > 150$ g C m$^{-2}$) was identified, which is confined to the water area of the Medvezhinsky–Spitsbergen shallow water and amounts to $152 \pm 36$ g C m$^{-2}$, with maximum values up to 190 g C m$^{-2}$, although the $NCP_N$ values in this area are not the highest ($58 \pm 9$ g C m$^{-2}$).
- Areas with relatively high values of primary production ($NCP_{Si} > 100$ g C m$^{-2}$) include the area in the southwestern part of the sea, the area in the southeast of the sea, as well as the area adjacent to the Franz Josef Land archipelago from the south. The highest $NCP_N$ values are observed in the southwestern part of the sea and amount to $73 \pm 7$ g C m$^{-2}$ with an average $NCP_{Si}$ value of $103 \pm 11$ g C m$^{-2}$.
- Areas with relatively low PP values ($NCP_{Si} < 100$ g C m$^{-2}$) include areas on the northern border of the sea between the Spitsbergen archipelagos and Franz Josef Land and the area adjacent to the coast of the Kola Peninsula and extending in the west from the Varanger fjord to the throat of the White sea.

The isotope parameter $\delta^{18}O$ ($\delta D$) has unique conservative properties. It does not depend on chemical–biological processes and is an ideal tracer for balance estimates. The use of this parameter makes it possible to solve one of the main problems in the assessment of primary production using changes in the concentration of the biogenic element—to determine the starting point, i.e., the maximum concentration of a nutrient before the photosynthesis process starts. Moreover, this level of concentration can change during the year when one water is replaced by another, which can also be taken into account when using this parameter. For example, for the Kara and Laptev Seas, where the volume of river runoff occupies a very large part, and the difference in nutrients concentrations is large (for example, silicon concentrations), it is necessary to constantly evaluate the maximum concentration of a nutrients. The nutrient supply decreases when one water changes to another.

The same state of affairs also applies to the content of meltwater in the surface layer, and the content of nutrients which is much less. If the initial concentrations are underestimated due to the influence of meltwater, then it can lead to overestimations of their consumption, and, consequently, to overestimations of primary production. Determination of the amount of nutrient consumption, i.e., the difference between the reference point and the measured value of nutrient concentrations, allows moving from average stoichiometric ratios to veritable ratios for a given point of research and a given period of the year. Usually the stoichiometric Redfield–Richards ratio is used, which in molar form is C:Si:N:P = 106:23:16:1, which is 41.1:20.9:7.2:1 by mass.

The average stoichiometric ratios calculated from the results of this study (by mass) for the entire Barents Sea in the euphotic layer during the period of maximum nutrient consumption (July–September) are reported in Table 5:

**Table 5.** Nutrient ratios and their content in different area and species (by mass).

| Ratio | Redfield–Richards [13,14] | Average at the Barents Sea * (Present Study) | Diatom ** | Peridinium ** |
|-------|---------------------------|----------------------------------------------|-----------|---------------|
| C:P | 41.1 | $56.6 \pm 3,3$ | 37.0 | 58.8 |
| Si:P | 20.9 | $13.2 \pm 0.9$ | 34.4 | 3.8 |
| Si:N | 2.9 | $1.4 \pm 0.8$ | 5.1 | 0.4 |
| N:P | 7.2 | $10.1 \pm 4.9$ | 6.7 | 8.1 |

Note: * calculated by Equation (13); ** [13,26] (Table 2).

Based on the results obtained from the measured values, the stoichiometric mass ratio for the Barents Sea can be written as follows: C:Si:N:P = 56.6:13.2:10.1:1 or in the atomic form 146:15:22:1.

In addition, it becomes possible to evaluate the change in stoichiometric ratios for the studied water area (Figure 14).

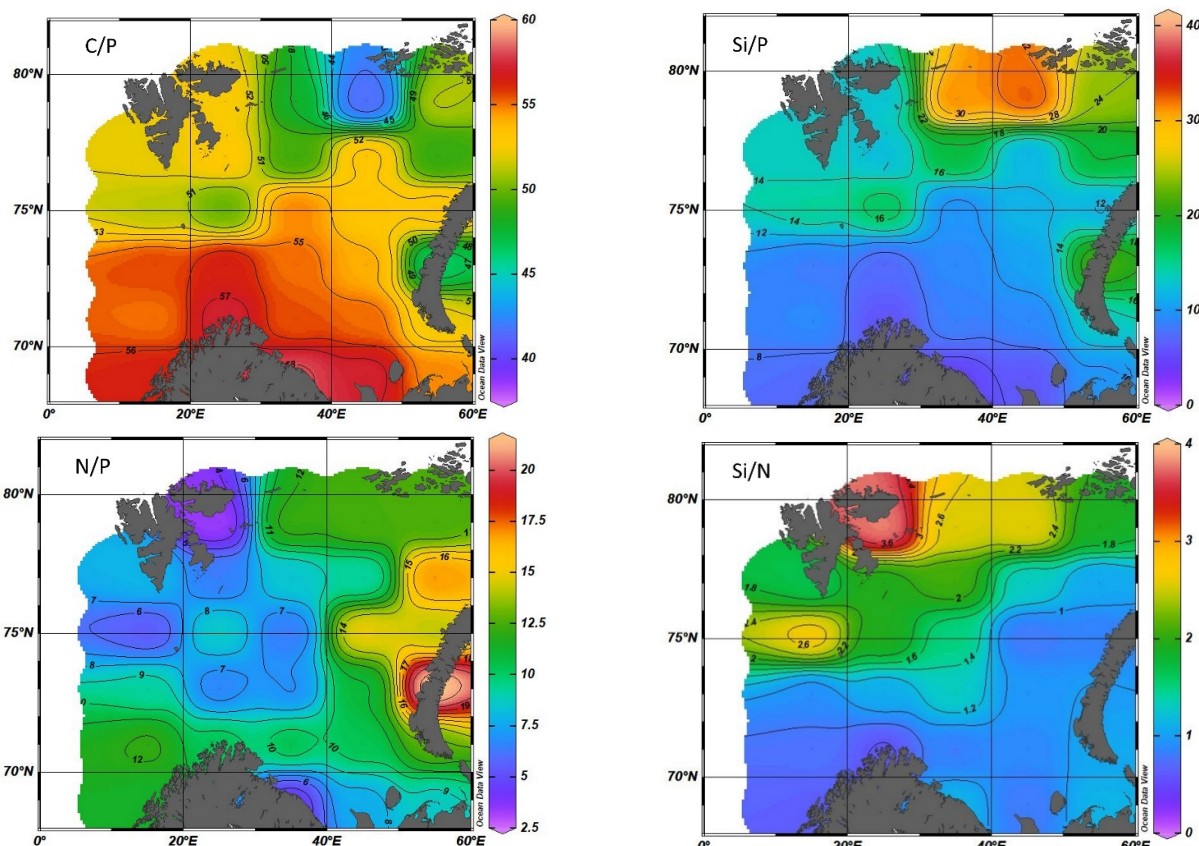

**Figure 14.** Distribution of stoichiometric ratios (by mass) in the Barents Sea during the period of maximum nutrient consumption (July–September).

The work [7] shows the change in the stoichiometric ratio in the Arctic Ocean. For the Barents Sea, the ratio POC:PON (particulate organic carbon (POC) and nitrogen (PON)) in atomic form was $8.5 \pm 1.8$, while the Redfield–Richards ratio was 6.62. However, it cannot be constant, because the classical ratio is an average ratio, and the real ratio is determined by the specific composition of phytoplankton, the formation of which consumes a different amount of nutrients. The average C:N ratio in the Barents Sea is $7.97 \pm 0.14$, which is quite close to the values presented in [7].

The technique described in this paper makes it possible to trace the change in both stoichiometric ratios and changes in primary production in various water areas, as well as in the annual cycle.

Since it was shown above that 94% of the Barents Sea phytoplankton consists of diatoms and dinoflagellates, then, consequently, the stoichiometric ratios should also vary in the range between the ratios for these two systematic groups. This is observed from the presented results with a shift of values from the ratios Redfield–Richards toward dinoflagellates. The C:P and C:N ratios in the Barents Sea are shifted by 29% and 23% accordingly. The N:P ratio falls out of this pattern in the Barents Sea, the value of which is outside the range of this value between diatoms and dinoflagellates by 25% which is still difficult

to explain and requires further research. Perhaps this is due to the need to revise the values of the content of biogenic elements in various systematic groups of phytoplankton.

The presented technique using the isotope parameter currently has its limitations. For its application, it is necessary to determine the average values of the isotope parameter, as well as the concentrations of nutrients for the base waters—river runoff and incoming ocean waters. If for shelf seas, these parameters can be determined from long-term observations of the values themselves, then for the scale of the ocean, for example, the Arctic Ocean, determining the initial values for such calculations is a separate task. For incoming Atlantic and Pacific waters, the values of the isotopic parameter and the concentration of nutrients can be determined using existing databases. In the case of river runoff, it is quite difficult to determine the values of these parameters for water areas outside the shelf seas, since, due to the huge catchment area of the inflowing rivers and the dispersion of rivers along the longitude and latitude, the dispersion of these values will be very large.

The presented work is based on the calculated values of the isotopic parameter $\delta^{18}O$ according to the dependence of this value on salinity, which has a correlation coefficient close to 1. However, the data for constructing this dependence for the Barents Sea, for the most part, are confined either to the waters surrounding the Spitsbergen archipelago or to the southeastern part of the sea. It is necessary to supplement the existing salinity–$\delta^{18}O$ series with new values. When assessing the productivity of some limited areas, to increase the accuracy of calculations, it is necessary to use unreconstructed $\delta^{18}O$ data but measure directly with parallel determinations of salinity and nutrients.

In addition, the calculations presented are based on the values of nutrient content in plankton, which were obtained and published in 1942. It is now necessary to revise and refine these values, including not only diatoms and dinoflagellates, but also other systematic groups of phytoplankton.

Moreover, at small values of $Q_{phyto}^n$ at the beginning of the photosynthetic activity season, in some areas, sum values of $d$ and p were observed greater than 1 or took negative values, which cannot be according to the conditions of the presented methodology. In the case where such values ($d > 1$ or $p > 1$ or $d < 1$ or $p < 1$) were received, the result was excluded from further calculations.

This can be caused by several reasons:

First, for small values of $Q_{phyto}^n$, the magnitude of the calculation error, relative to the obtained value of $Q_{phyto}^n$ is very large;

Secondly, the sum of diatom and peridinium plankton biomass is not 100% of the total biomass. In some years, *Phaeocystis pouchetii* (class Prymnesiophyceae) can reach high development levels at the peak of spring blooming, giving up to 95% of the total phytocenosis biomass. That can change the ratio of nutrient consumption values in such areas.

## 5. Conclusions

The isotope parameter $\delta^{18}O$ ($\delta D$) has unique conservative properties. It does not depend on chemical–biological processes and is an ideal tracer for balance estimates. The combination of the use of isotopic parameters and nutrients has great prospects, which makes it possible to study a part of the marine ecosystem, including hydrological–hydrochemical–hydrobiological (phytoplankton) processes, as a single system of their relationship.

Verification of the presented method by comparing the relative values of diatoms and dinoflagellates' contribution to the total production (for the case of calculated values), on the one hand, and to the total biomass in the case of its direct measurement, on the other hand, gave fairly similar results. In addition, a fairly good convergence of primary production values was obtained by comparing with the results of model calculations published by other teams of authors for the water area surrounding the Spitsbergen archipelago.

Based on the presented methodology, the distribution of stoichiometric ratios in the Barents Sea is shown for the first time. The features of nutrient consumption and changes in primary production in the Barents Sea were studied.

According to the highest values of primary production in the summer–early autumn period in the Barents Sea, an area with maximum values of this parameter (GPP > 150 g C m$^{-2}$), three areas with increased values (GPP > 100 g C m$^{-2}$), and two areas with relatively low values (GPP < 100 g C m$^{-2}$) were identified. The region with the maximum values of primary production is confined to the waters surrounding the Medvezhinsky–Spitsbergen shallow water.

In the eastern part of the sea in the summer, in the south from the mainland coast to the archipelago of Franz Josef Land, almost complete consumption of nitrogen is observed, which is a limiting factor in the development of the photosynthesis process. In the western part of the sea, the values of nitrogen consumption are much lower and are in the range from 40 to 70%, although the values of primary production are maximum here. In the northern part of the sea, between the Svalbard and Franz Josef Land archipelagos, silicon can be a limiting factor in the development of the photosynthesis process.

The use of this technique in the presence of a long series of salinity and nutrients allows us to proceed to the study of the climatic variability of these parameters, starting from the variability of the consumption of nutrients and ending with the variability of the productivity of the studied water area.

The advantage of this approach is that the methods of chemical analysis for the determination of nutrients are quite simple, unified in different countries, and well-developed for their use in the field. In addition, rather large databases containing these values in various areas of the World Ocean have accumulated.

Previous works [45] have shown that there is a relationship between the isotope parameter $\delta^{18}$O and salinity with a correlation coefficient close to 1, so the value of $\delta^{18}$O can be restored for all salinity values, as studied and proved in the section "Materials and Methods". This combination of isotopic parameters application and nutrients has great prospects.

Firstly, this makes it possible to study a part of the marine ecosystem, including hydrology–hydrochemistry–hydrobiology (phytoplankton), as a single system of interconnection of the studied parameters.

Secondly, the presence of long series of salinity and nutrients makes it possible to proceed to observations of the climatic variability of these environmental parameters starting from nutrients consumption variability to the productivity variability within research waters.

**Author Contributions:** Conceptualization, A.A.N.; methodology, A.A.N.; formal analysis, E.I.D., P.R.M. and I.A.P., investigation, E.I.D., P.R.M. and I.A.P.; data curation, E.I.D., P.R.M. and A.A.N.; visualization, A.A.N.; writing—original draft preparation, A.A.N. and P.R.M.; writing—review and editing, A.A.N., P.R.M. and E.I.D.; funding acquisition, P.R.M. All authors have read and agreed to the published version of the manuscript.

**Funding:** This study was funded by the Ministry of Science and Higher Education of the Russian Federation. State assignment number FMEE-2021-0029 (0188-2021-0029), theme "Planktonic communities of the Arctic seas under the conditions of modern climatic changes and anthropogenic impact".

**Data Availability Statement:** NODC WOA: https://www.ncei.noaa.gov/products/world-ocean-atlas (accessed on 23 October 2022); NASA: https://data.giss.nasa.gov/o18data/ (accessed on 25 September 2022)

**Acknowledgments:** We thank the personnel of the Administration and Plankton Laboratory at the Murmansk Marine Biological Institute for their aid in the collection and processing of water samples. We would like to thank the scientists and staff who created and maintain the NODC and NASA databases. We would like to thank the developers of the ODV software package, that allowed the accumulation of and rapid processing of the oceanographic data. Thanks to reviewers for constructive suggestions that helped to considerably improve the manuscript.

**Conflicts of Interest:** The authors declare no conflict of interest.

## Appendix A

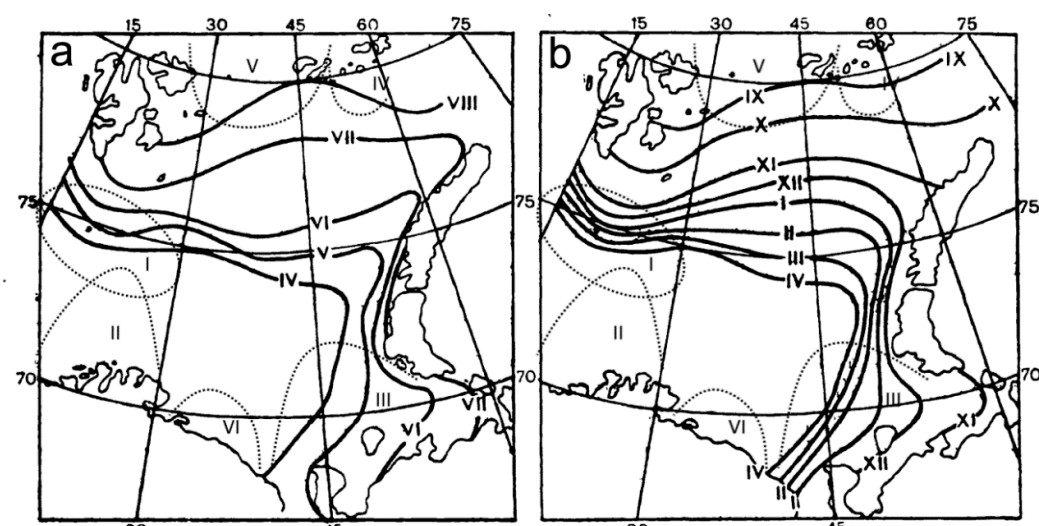

**Figure A1.** Average long-term position of ice edges in the Barents Sea ((**a**), April to August; (**b**) September to April, according to [25]). Dashed lines—area boundaries.

**Table A1.** The calculated regression line, the used statistical test and the significance level.

| Parameters | Type of regression | Serie 2 |
| --- | --- | --- |
| Number of members | | 1062 |
| $R^2$ | $\delta^{18}$O-Sal | 0.924 [1] |
| $R^2$ | $\delta^{18}$Omeasure-$\delta^{18}$Ocalc | 0.894 [2] |
| *t*-statistics | $\delta^{18}$Omeasure-$\delta^{18}$Ocalc | 0.036 [2] |
| *t*-tabl ($p = 0.05$) | 1062 (Number of members) | 1.64 |
| f-statistics | $\delta^{18}$Omeasure-$\delta^{18}$Ocalc | 8.44 [2] |
| f-tabl ($p = 0.05$) | 1062 (Number of members) | 3.84 |
| f-tabl ($p = 0.01$) | 1062 (Number of members) | 6.64 |

Note: [1] coefficient of determination of the relationship between $\delta^{18}$O and salinity in Serie 1. [2] statistics between calculated and measured values for Serie 2.

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
