# Peer review of "Parameter δ18O in the Marine Environment Ecosystem Studies on the Example of the Barents Sea"

_water, doi:10.3390/w15020328_

Round 1
Reviewer 1 Report
The article discusses a useful modification of the method for calculating a nutrients concentration, nutrients consumption and a conservative concentration, using data on the isotopic composition of oxygen and hydrogen in water. The presented approach is very promising, given that at present it is possible to determine the necessary parameters directly during navigation. Therefore, the article should undoubtedly be accepted for publication. However, there are a number of comments to the text presented. The most general remarks are set out below, while specific corrections and questions can be found in the text of the attached (see file “water-2018778-peer-review-v1+rev-02.pdf”). The article requires significant processing in terms of design, followed by re-submission to the reviewer.
The calculations used estimates of the average for a number of quantities with excessive accuracy (for example, the isotopic composition of oxygen is given with an accuracy of three digits after the decimal point).
Regularly the extremely complex grammatical constructions are used, which makes it very difficult to understand the authors' arguments. A large number of minor editorial comments (they are noted in the pdf-text) must be taken into account when finalizing the article.
Some data are given without references to sources.
Some of the calculations contradict the generally accepted ratios.
For example, "The N:P ratio falls out of this pattern in the Barents Sea, the value of which is outside the range of this value between diatoms and dinoflagellates by 25%" (lines 832-833).
However, no hypothesis regarding the detected deviation is given.
In Lines 789-792 is discussed an influence on thawed water on estimation of initial nutrient concentration. But everywhere above it is said about the transformation of water into ice (that is, the freezing of water, and not the melting of ice). The terminology needs to be unified to avoid the contradiction arises.
Apparently, it is necessary to structure the text a little differently.
For example:
– Section “4.2. Perspective” should be moved to the Conclusion;
– Section “4.1. Limitations of the work” should be moved to discussion.
– Section “3.3. Experimental conclusions" has an unfortunate title. It is better to move this piece of text to the "Discussion" section.

Author Response
Dear experts and editorial staff of Water journal!
Here is an edited version of our manuscript.
Thank you very much for giving us an opportunity to publish an article in your journal. We also liked to thanks the Experts for their work and detailed review of our manuscript. We tried to carefully consider all the comments and make changes according to Experts comments. Undoubtedly, all comments and suggestions of experts, and corrections we made improved the quality of the manuscript.
Most of the comments and suggestions are taken into consideration, but some of them requires detailed response. We organized all the answers and comments into one table (separately for each Reviewer).
Thank you for your time.
Best wishes,
team of authors
Corresponding author
Namyatov Alexey, PhD
leading researcher of
Murmansk Marine Biological Institute

Reviewer 2 Report
Dear Authors,
the paper is interesting but it needs some work before publications.
I write all my suggestions in the attached file. It is also important to check character font and size that has to be the same in the manuscript and to verify the good quality of the figures. In many of the it has been used the diciture left and right but I suggest you to use labels in both figure and relative caption.
I hope that my revision work will improve your manuscript.

Author Response

(The authors gave the same response as above.)
